

# Role of climate model dynamics in estimated climate responses to anthropogenic aerosols

Kalle Nordling[1], Hannele Korhonen[1], Petri Räisänen[1], Muzaffer Ege Alper[1], Petteri Uotila[2], Declan O'Donnell[1], and Joonas Merikanto[1]

[1]Finnish Meteorological Institute, Helsinki, Finland
[2]INAR/Physics, University of Helsinki, Helsinki, Finland

**Correspondence:** Kalle Nordling (kalle.nordling@fmi.fi)

**Abstract.** Significant discrepancies remain in estimates of climate impacts of anthropogenic aerosols between different general circulation models (GCMs). Here, we demonstrate that eliminating differences in model aerosol or radiative forcing fields results in close agreement in simulated globally averaged temperature and precipitation responses in the studied GCMs. However, it does not erase the differences in regional responses. We carry out experiments of equilibrium climate response to modern day anthropogenic aerosols using an identical representation of anthropogenic aerosol optical properties and aerosol-cloud interactions, MACv2-SP, in two independent climate models (NorESM and ECHAM6). We find consistent global average temperature responses of $-0.48(\pm 0.02)$ K and $-0.50(\pm 0.03)$ K and precipitation responses of $-1.69(\pm 0.04)\%$ and $-1.79(\pm 0.05)\%$ in NorESM1 and ECHAM6, respectively, compared to modern-day equilibrium climate without anthropogenic aerosols. However, significant differences remain between the two GCMs regional temperature responses around the Arctic circle and the equator and precipitation responses in the tropics. The scatter in the simulated globally averaged responses is small in magnitude when compared against literature data from modern GCMs using model intrinsic aerosols but same aerosol emissions $(-(0.5-1.1)$ K and $-(1.5-3.1)\%$ for temperature and precipitation, respectively). The Pearson correlation of regional temperature (precipitation) response in these literature model experiments with intrinsic aerosols is 0.79 (0.34). The corresponding correlation coefficients for NorESM1 and ECHAM6 runs with identical aerosols are 0.78 (0.41). The lack of improvement in correlation coefficients between models with identical aerosols and models with intrinsic aerosols implies that the spatial distribution of regional climate responses is not improved via homogenizing the aerosol descriptions in the models. Rather, differences in the atmospheric dynamic and high latitude cloud and snow/sea ice cover responses dominate the differences in regional climate responses. Hence, further improvements in the model aerosol descriptions can be expected to have a limited value in improving our understanding of regional aerosol climate impacts, unless the dynamical cores of the climate models are improved as well.





## 1 Introduction

Making reliable predictions on future changes in regional climates is crucial for estimating how climate change will impact people and societies (Hawkins et al., 2016), but there are still large uncertainties related to climate change predictions on regional scales (Giorgi and Francisco, 2000; Feser et al., 2011). Anthropogenic aerosol particles can be an important driver

for regional climate change due to the near-instantaneous response of local aerosol concentrations to changes in emissions, their direct radiative properties, and their ability to modify cloud microphysical processes. However, reliable implementation of aerosol effects into global climate models has been challenging. Several aerosol processes are still not well-understood (Boucher et al., 2013), and there exists an enormous scale difference between the microphysical processes and the resolution of global scale models (Carslaw et al., 2013).

Varying descriptions of aerosols and aerosol-cloud interactions cause a widespread in aerosol radiative forcing and climate impacts between different GCMs (Wilcox et al., 2015). Shindell et al. (2015) compared historical CMIP5 runs with and without anthropogenic forcing from aerosols, ozone, and land use. The forcing showed a very large spatial variation with globally averaged values that ranged between $0.15\,\mathrm{Wm^{-2}}$ and $-1.44\,\mathrm{Wm^{-2}}$ (the aerosol contribution being between $-0.29\,\mathrm{Wm^{-2}}$ and $-1.44\,\mathrm{Wm^{-2}}$). The combined changes in aerosol, ozone and land use produced globally averaged transient temperature

responses between $0.00\,\mathrm{K}$ and $-1.33\,\mathrm{K}$ over the twentieth century, with the spatial pattern of the temperature response varying significantly between the models. Overall, the inclusion of aerosols in CMIP5 models nevertheless improved the historical temperature trends compared to observations. This applied particularly to models including sophisticated parameterizations for aerosol cloud droplet activation (Ekman, 2014).

Besides reducing the global temperature, anthropogenic aerosols are also known to reduce global precipitation (Ramanathan,

2005) and to significantly modify the Asian monsoon (Bollasina et al., 2011; Salzmann et al., 2014) . Wang (2015) demonstrated that among CMIP5 models the changes in anthropogenic aerosols dominated the total precipitation changes from the pre-industrial era to the present day. Most of this change was caused by the remote impact of aerosols rather than by direct effects on local cloud processes in all but heavily aerosol-loaded regions, such as in the Indian monsoon region. Also for precipitation changes, an improved representation of aerosol-cloud interactions was found to be the key factor in reproducing

consistent distributions of past precipitation change.

Improvements in model aerosol descriptions have not succeeded to remove the large uncertainty in aerosol climate effects. After CMIP5, the most representative multi-model results on aerosol climate impacts have been provided by Samset et al. (2018). They compared the equilibrium climate responses for complete removals of model intrinsic anthropogenic aerosols among four state-of-art fully coupled climate models, with aerosol emissions from CMIP5 (Lamarque et al., 2010). In their

study, removing the aerosols produced global-mean temperature increases between 0.5 and 1.1 K and precipitation increases between 1.5and 3.1%. In another recent study, Kasoar et al. (2016) reduced anthropogenic $SO_2$ emissions from China in three independent climate models. There, identical emission reductions lead to simulated changes in aerosol optical depth and shortwave radiative flux over China that varied by up to a factor of 6 between the models. The three models also exhibited large differences in their global and regional temperature responses. However, it is unclear to which degree the existing spread



in aerosol climate impacts among current climate models results from differences in modeled aerosols or from differences in model dynamic responses to aerosols. Only standardized aerosol perturbations across different models can entangle these sources of uncertainties in aerosol climate effects (Stier et al., 2013).

Here, we explore how robust the aerosol climate response would be in modern GCMs if the anthropogenic aerosols and their
cloud interactions could be modeled exactly. To assess this question we carry out long equilibrium climate simulations with fixed greenhouse gas concentrations and prescribed aerosol fields using the MACv2-SP aerosol description (Stevens et al., 2017) in two modern GCMs, NorESM and ECHAM6. The MACv2-SP is partly based on observational data and provides a simple representation of global aerosol optical properties. It also includes a simple empirical fit for aerosol-cloud-albedo effects. These experiments allow us to single out the contribution of climate model dynamics to the intermodel differences in
the response to anthropogenic aerosols. We will compare our results against the dataset by Samset et al. (2018) to investigate the robustness of global and regional climate responses in modern climate models using interactive or prescribed aerosols.

## 2 Methods

### 2.1 Applied climate models and set-up

We carry out modern day equilibrium climate simulations with two independent climate models, ECHAM6.1 and NorESM1.
ECHAM6.1 (Stevens et al., 2013) is the sixth generation of ECHAM general circulation model developed in Max Planck Institute with 47 sigma hybrid vertical levels, with the model top at 0.01 hPa and a horizontal resolution of $1.9° \times 1.9°$. Original ECHAM model branched from an early version of the European Centre for Medium-Range Weather Forecasts (ECMWF) model for climate studies. NorESM1 is the Norwegian Earth system model with 26 sigma hybrid vertical levels (highest model level at 2.9 hPa) and $1.9° \times 2.5°$ horizontal resolution (Bentsen et al., 2013; Iversen et al., 2013; Kirkevåg et al., 2013).
NorESM1 is based on the CCSM4 model operated at NCAR. Thus, the two models applied in our study do not share a common development history. Here, both models were ran with identical fixed modern-day greenhouse gas concentrations. Oceans were simulated with the intrinsic slab ocean configurations of the models. This idealization removes the effect of natural and aerosol induced variations in ocean dynamics and restricts our study to the response in atmosphere/sea ice dynamics only.

### 2.2 Standardized aerosol representation

MACv2-SP is a standardized representation of anthropogenic aerosol radiative effects, accounting for the direct radiative as well as the cloud albedo effect of anthropogenic aerosol (Stevens et al., 2017). However, cloud lifetime effect is not taken into account. Anthropogenic aerosols are represented by nine 3D time-varying Gaussian plumes defining the aerosol optical depth, single scattering albedo and asymmetry parameter. Four of these plumes represents aerosol emissions from biomass burning, the other five are associated with industrial emissions. The industrial plumes originate from Europe, North America,
East Asia, South Asia and Australia and the biomass plumes from North Africa, South America, South central Africa and Maritime Continent (Fig. 1 and Table 1 in Stevens et al. (2017)). The plumes differ in their annual cycle and optical properties,



and have a realistic horizontal and vertical structure that represents the transports of aerosols with prevailing winds. The aerosol properties of are based on aerosol climatology by (Kinne et al., 2013), derived from ground-based sun-photometer networks (AERONET) merged onto background maps from global models participating in the Aerosol Model Intercomparison Project (AeroCom). The cloud albedo effect is parameterized by modifying the model-intrinsic natural cloud droplet number

concentration (CDNC). The relation between aerosol optical depth and CDNC is derived from MODIS data. MACv2-SP allows for a simple and observation-based representation of the changes in aerosol optical properties and cloud droplet number concentrations due to anthropogenic aerosols.

## 2.3 Model experiments and analysis

Sets of 100-year equilibrium climate runs for the year 2005 were conducted with both models, with the last 60 years used for

the analysis: (1) The control run (CTRL) included only natural aerosols, and was constructed from two identical runs with small initial condition perturbations; (2) The MACSP run included both natural and anthropogenic aerosols for the year 2005. In addition, for NorESM1, a third run EF was carried out. This run employed the time-varying 3D aerosol radiative forcing field computed from the ECHAM6's MACSP run. A summary of the runs is given in Table 1.

**Table 1.** Summary of the performed model runs

| Runs | Forcing | Models |
|------|---------|--------|
| CTRL | natural aerosols | ECHAM6, NorESM1 |
| MACSP | MACv2-SP + natural aerosols | ECHAM6, NorESM1 |
| EF | Forcing field from ECHAM6 | NorESM1 |

Based on these runs, the following three experiments were defined to estimate the effect of anthropogenic aerosols: ECHAM6-

MACSP (the difference between the MACSP and CTRL runs for ECHAM6), NorESM1-MACSP (MACSP minus CTRL for NorESM1), and NorESM1-EF (EF minus CTRL for NorESM1). The analysis of the results was based on monthly-mean values of data, and focused on the effects of MACv2-SP aerosols on near-surface temperature, precipitation, surface albedo and total cloud cover. The statistical significance of the responses was evaluated using Student's t-test with an auto-correlation correction according to Zwiers and von Storch (1995). The response uncertainties in global mean values were estimated by the

standard error of means taking into account lag-1 auto-correlation according to Zwiers and von Storch (1995). The instantaneous radiative forcing was calculated using double radiation calls with and without MACv2-SP aerosols during the slab ocean runs.



## 3 Results

### 3.1 Aerosol radiative forcing

The total radiative forcing from MACv2-SP anthropogenic aerosol description was found to be very similar for the two models (see Fig. 1). For ECHAM6, the MACv2-SP aerosol scheme produces a $-0.64\,\mathrm{Wm^{-2}}$ global average total shortwave radia-

tive forcing at the top of the atmosphere (TOA) for year 2005, with $-0.35\,\mathrm{Wm^{-2}}$ arising from direct and $-0.29\,\mathrm{Wm^{-2}}$ from indirect radiative forcing. For NorESM1, the same aerosol scheme produces a slightly higher global radiative forcing of $-0.69\,\mathrm{Wm^{-2}}$ at TOA, with $-0.36\,\mathrm{Wm^{-2}}$ direct and $-0.33\,\mathrm{Wm^{-2}}$ indirect radiative forcing. Figure A1 shows the maps of aerosol direct and indirect radiative forcings in the two models as calculated here. The largest difference in the total forcing was found over South East Asia ($3.20\,\mathrm{Wm^{-2}}$), where also the largest absolute forcing was found in both models. Fiedler et

al. (2018) have calculated both the MACv2-SP effective radiative forcing as well as the instantaneous radiative forcing using double radiation calls with fixed sea surface temperature for the two climate models used here. They showed that with fixed sea surface temperature the MACv2-SP aerosols produce an instantaneous radiative forcing of $-0.60\,\mathrm{Wm^{-2}}$ and $-0.68\,\mathrm{Wm^{-2}}$ in ECHAM6 and NorESM1, respectively. The correlation coefficient for the regional total forcing in the two models due to MACv2-SP is $0.97$, and $0.90$ for direct and $0.89$ indirect forcings only. Thus, the regional differences in direct and indirect

forcing somewhat compensate for each other.

We used a Gaussian process emulation technique (O'Hagan, 2006) to assess the causes for the regional differences in aerosol radiative forcing (see Appendix B for details). Our analysis showed that differences in cloud cover and surface albedo can explain nearly all of the variance in total instantaneous shortwave radiation between ECHAM6 and NorESM1. Our sensitivity analysis reveals that in the regions with largest radiative forcing (close to the center of the MACv2-SP plumes) the difference

in model cloud cover dominates the difference in model shortwave forcing. Vice versa, in regions with low aerosol radiative forcing the differences in surface albedo dominates the differences in forcing. We note that these results apply only to fixed aerosol fields produced by the MACv2-SP representation. Previous research shows that the aerosol radiative forcing can also depend on the meteorology (surface winds and precipitation) produced by the models, partly driven by the natural variability of the climate system (Baker et al., 2015; Bony et al., 2015; Shepherd, 2014).

### 3.2 Climate response to the addition of anthropogenic aerosols

#### 3.2.1 Temperature

We obtain a robust global temperature response of $-0.5$ K due to the inclusion of MACv2-SP anthropogenic aerosols in both models. For ECHAM6-MACSP experiment the global mean near-surface temperature response is $-0.50(\pm0.03)$ K, with regional values ranging from $+0.30$ K to $-2.10$ K. For NorESM1-MACSP experiment the global mean value is $-0.48(\pm0.02)$

K and the regional values range between $+0.39$ K and $-2.28$ K.

Figure 2 shows the regional temperature response to the inclusion of anthropogenic MACv2-SP aerosols. The spatial correlation between ECHAM6-MACSP and NorESM1-MACSP experiments is $0.81$ for full experiments with $60+120$ years of





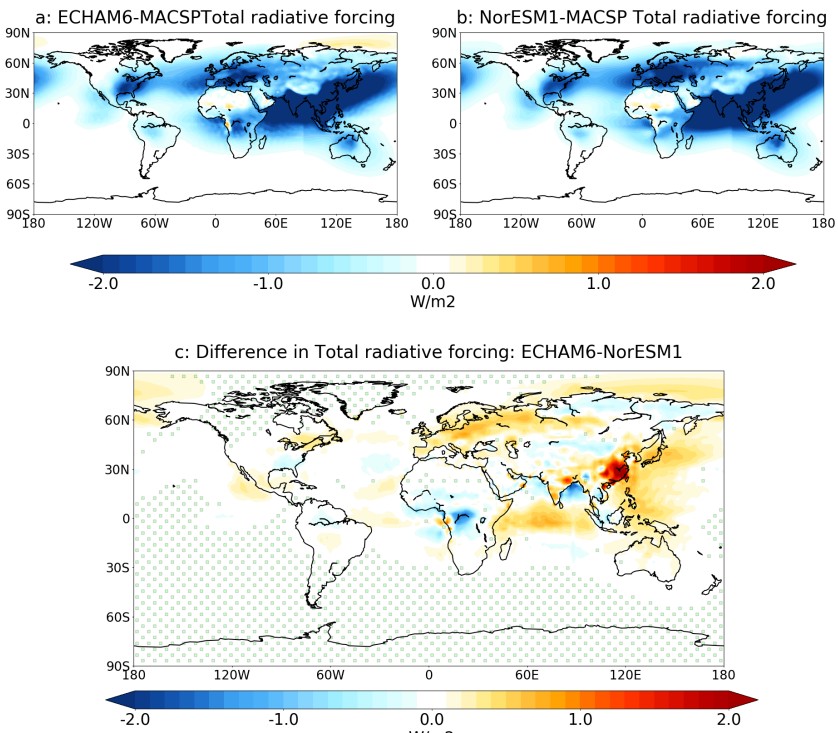

**Figure 1.** The total radiative forcing at top of the atmosphere produced by MACv2-SP aerosols. The top left figure shows the forcing in ECHAM6-MACSP experiment and the top right figure in NorESM1-MACSP experiment. Below is shown the difference in the forcing between the two models. Small green circles mask the areas where results are not statistically significant at the p level $< 0.05$.

MACSP and CTRL runs in both models. Largest cooling in ECHAM6 is located in Southeast Asia whereas in NorESM1 the largest cooling is found near the Russian Far East and north of Japan, with a second minimum over the Greenland sea. Small positive temperature responses are found close to the Antarctic coast in both models, but these temperature responses are not statistically significant and are related to natural variations in sea ice. We found some significant correlation between
5   the regional aerosol forcing and regional temperature response in both models: $0.39$ in ECHAM6 and $0.29$ in NorESM1, respectively. Among the CMIP5 model considered in Shindell et al. (2015), the multimodel mean regional correlation between the combined effective aerosol and ozone forcing and temperature response was slightly negative ($-0.1$), varying between negative values in some models and positive values among others.

    Figure 3a shows the zonal-mean temperature responses obtained from ECHAM6-MACSP and NorESM1-MACSP experi-
10   ments. These experiments show a moderate cooling due to anthropogenic aerosols across the southern hemisphere latitudes,





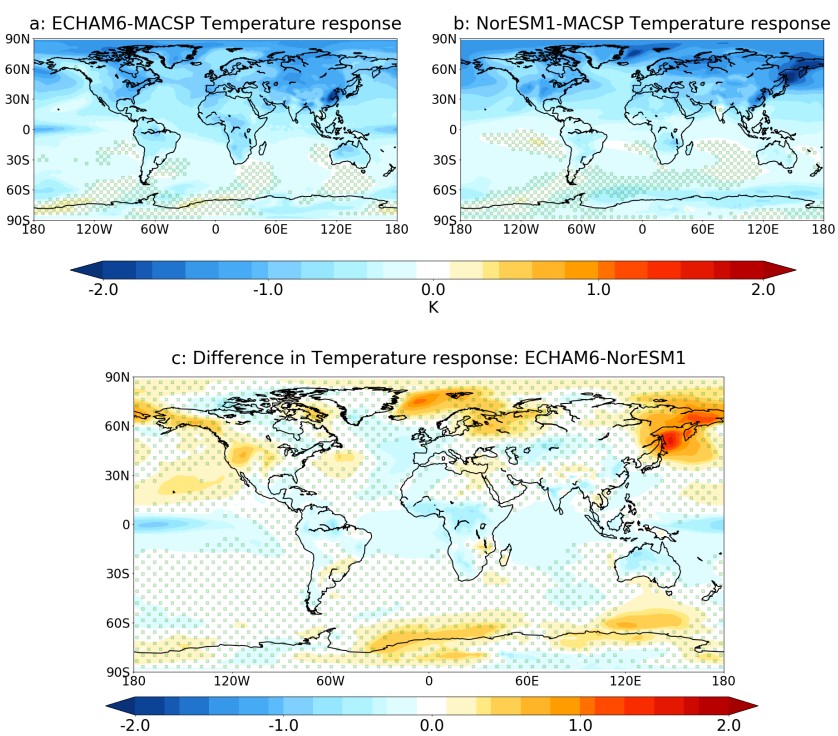

**Figure 2.** Near surface temperature response to the addition of anthropogenic (MACv2-SP) aerosols. The top left figure shows the response for ECHAM6-MACSP experiment and the top right figure for NorESM1-MACSP experiment. Below is shown the difference in the responses between the two models. Small green circles mask the areas where results are not statistically significant at the p level < 0.05.

whereas in the northern hemisphere the cooling response clearly strengthens towards the high latitudes. The modeled regional temperature responses between ECHAM6 and NorESM1 simulations disagree the most in high latitude regions as seen in Figure 2c, also associated with largest differences in surface albedo feedback (snow/sea ice) between the models (see Figure A2 ). This feedback, together with ocean circulation feedback, also dominates the regional differences in temperature responses
5 to homogeneous greenhouse gas forcing among different climate models (Shindell et al., 2015). The similarity in zonal-mean temperature response at high northern latitudes in ECHAM and NorESM1 is curious, as NorESM1 shows a more positive surface albedo response (Fig. A2) and a more negative cloud cover response (Fig. A3), both of which should favor stronger cooling at high latitudes.





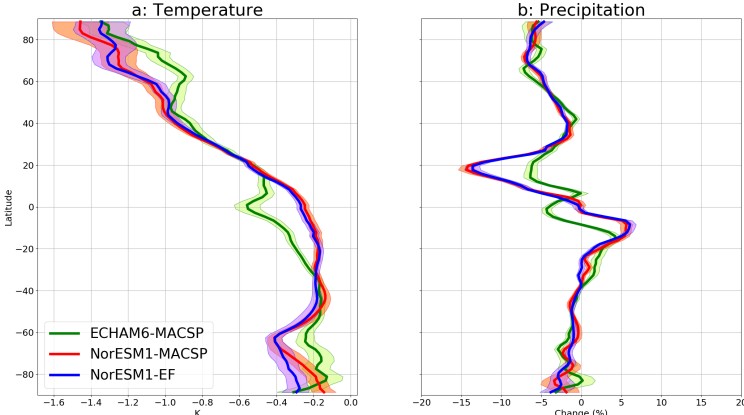

**Figure 3.** Impact of MACSP anthropogenic aerosols on zonal-mean temperature (K) and precipitation (%) in ECHAM6-MACSP, NorESM1-MACSP and NorESM1-EF experiments. The shaded area shows the standard error of mean as a function of latitude.

**Table 2.** Summary of global mean change of temperature and precipitation due to modern day. Standard error of means are shown in brackets

|  | Near surface temperature | Precipitation (%) |
|---|---|---|
| ECHAM6-MACSP | -0.50 ($\pm$ 0.03) | -1.79 ($\pm$ 0.05) |
| NorESM1-MACSP | -0.48 ($\pm$ 0.02) | -1.69 ($\pm$ 0.04) |
| NoreESM-EF | -0.49 ($\pm$ 0.01) | -1.82 ($\pm$ 0.04) |

### 3.2.2 Precipitation

The inclusion of anthropogenic aerosols results in a similar global reduction of precipitation in all experiments, with ECHAM6-MACSP showing a change of $-1.79 \pm 0.05$ % and NorESM1-MACSP change of $-1.69 \pm 0.04$ % in annual precipitation (Table 2). The regional changes of the precipitation patterns are shown in Figure 4. The spatial correlation between the precipitation responses in full ECHAM6-MACSP and NorESM1-MACSP experiments is $0.47$, which is much lower than the corresponding correlation for temperature. In addition, while the temperature responses are negative almost globally, both positive and negative responses occur for precipitation, with relatively sharp edges between regions with different signs of changes. While similar large-scale features of precipitation changes can be seen in both models, their dislocation leads to a weaker regional correlation than for the temperature response. In both models, the relative changes in the convective precipitation are larger than the relative changes in large-scale precipitation. Also consistently across the two models, the seasonal response in the total precipitation is similar, with the largest changes in June-July-August (see Table A1). Both models consistently show an overall drying of the Northern Hemisphere, with some statistically significant regional increase in precipitation over the North-West Arfica.



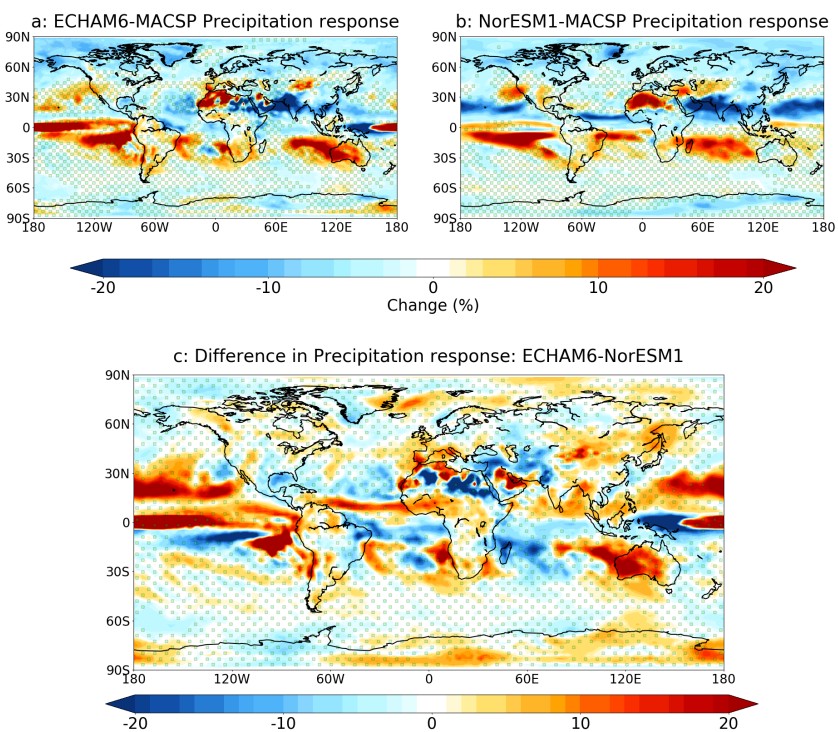

**Figure 4.** The upper left shows ECHAM6-MACSP experiment precipitation response to adding MACv2-SP aerosols and the upper right figure shows the same for NorESM1-MACSP experiment. Below is the intermodel difference in precipitation response. The green dots mark the regions where the MACv2-SP aerosols do not have a statistically significant impact at the level p < 0.05.

Both models show a maximum reduction in total precipitation around $15° - 20°$ N and a maximum increase around $10° - 15°$ S, associated with an asymmetric response in Hadley circulation across the equator (see Figs. 3b and 4). Changes in precipitation in the tropics are also related to changes in vertical motion in same region (see Fig. A4). This is suggestive of a southward shift of the Intertropical convergence zone (ITCZ) associated with a change in hemispheric temperature gradient

5  (Broccoli et al., 2006). The inclusion of anthropogenic aerosols results in decreased precipitation in the South Asian monsoon region (defined here as the land region over $5°–25°$N, $65°–110°$E) (Fig. 3). In June-August, the monsoon precipitation is decreased by 12.8% in ECHAM6-MACSP and 15.3% in NorESM1-MACSP experiments. Reduction of monsoon precipitation due to the anthropogenic aerosols has also been reported in several previous studies (Ganguly et al., 2012; Li et al., 2018; Polson et al., 2014; Bollasina et al., 2011). Opposite to the seasonal cycle in temperature response, the largest precipitation response

10  occurs in summer during the Asian monsoon season. The two models show a different response over the West African monsoon



region (5° S – 25° N, 20° W – 20° E), with NorESM1-MACSP experiment showing a statistically significant reduction in precipitation of −5.3 % while ECHAM6-MACSP experiment does not show a significant change (−1.8 %). In the vicinity of the Australian continent, ECHAM6-MACSP experiment shows an area of increased precipitation extending from the Indian Ocean to the Western Australia, while in NorESM1-MACSP experiment, the increase is located entirely over the Indian ocean.

There appear to be several causes for the differences in the precipitation response between the two models. For instance, there is a relationship between the difference of the regional precipitation response and the difference in vertical velocity response (correlation coefficient 0.44 between Figure 4c and Figure A4c). However, also the difference in the initial equilibrium state of precipitation patterns correlates weakly with the difference in the precipitation response (correlation coefficient 0.23). Furthermore, differences in the cloud cover responses are also related to differences in precipitation responses (correlation

coefficient 0.32). The vertical velocity correlates also with the total cloud cover response (correlation coefficient 0.41)

### 3.2.3    Comparison between the NorESM1-MACSP and NorESM1-EF experiments

We now briefly discuss the differences between the NorESM1-MACSP and NorESM1-EF experiments. As noted in Sect. 2.3, the difference between these experiments is that in NorESM1-MACSP, the radiative forcing due to the MACv2-SP aerosols is computed using NorESM1's own meteorology and own radiation scheme, while in NorESM1-EF, forcings from ECHAM6's

MACSP run are applied. The general finding here is that the results for these two experiments are very similar. The global-mean temperature response is $−0.48(\pm0.02)$ K for NorESM1-MACSP and $−0.49(\pm0.01)$ K for NorESM1-EF, while the global-mean precipitation responses are $−1.69(\pm0.04)$% and $−1.82(\pm0.04)$%. Also, the zonal-mean temperature and precipitation responses in NorESM1-MACSP and NorESM1-EF are very similar (Fig. 3). The spatial correlation in response between full NorESM1-MACSP and NorESM1-EF experiments is as high as 0.97 for temperature and 0.95 for precipitation, which are

much higher than the correlations between NorESM1-MACSP and ECHAM6-MACSP responses (0.81 and 0.47). Indeed, with the exception of the global-mean precipitation response, for which the ECHAM6-MACSP value (-1.79 $\pm$0.05%) falls between the NorESM1-MACSP and NorESM1-EF, the responses in the two NorESM1 experiments are closer to each other than the ECHAM6-MACSP response. Therefore, it can be concluded that the differences in the effects of MACv2-SP aerosols between ECHAM6 and NorESM1 are mainly related to differences in the model dynamical responses, not to the differences

in the aerosol forcing fields.

### 3.3    Comparison to models with interactive aerosols

Finally, we compare the obtained equilibrium temperature and precipitation responses with prescribed MACv2-SP aerosols in ECHAM6 and NorESM1 against those equilibrium climate responses from four fully coupled climate models (CESM1, GISS, HadGEMS2, and NorESM1) with intrinsic aerosol schemes but the same aerosol emissions, reported by Samset et al. (2018).

In the four models considered by Samset et al. (2018), the global average temperature responses were 0.5 K, 0.5 K, 1.1 K and 0.6 K, and precipitation responses 1.5%, 1.8%, 2.6% and 3.1%, respectively. We obtain similar temperature responses of $0.48 − 0.50$ K and precipitation responses of $1.69 − 1.82$ % using the prescribed MACv2-SP aerosol description.





**Table 3.** Intermodel correlations of regional temperature response for the Samset et al. (2018) models and our models. The average correlation coefficient between the Samset et al. (2018) models is 0.79 with a standard deviation of 0.05; the average correlation coefficient between the models used in this study and the Samset et al. (2018) models is 0.76. The correlations are calculated for 50 years with and 50 years without anthropogenic aerosols. Correlation for our whole dataset (60+120 years) is shown in brackets. The range is the standard deviation between results obtained for two different CTRL runs.

|  | CESM1 | GISS | HadGEM2 | NorESM1 | ECHAM6-MACSP | NorESM1-MACSP |
|---|---|---|---|---|---|---|
| GISS | 0.74 |  |  |  |  |  |
| HadGEMS2 | 0.83 | 0.79 |  |  |  |  |
| NorESM1 | 0.82 | 0.71 | 0.87 |  |  |  |
| This study |  |  |  |  |  |  |
| ECHAM6-MACSP | 0.75±0.01 | 0.72±0.02 | 0.75±0.01 | 0.74±0.02 |  |  |
| NorESM1-MACSP | 0.80±0.01 | 0.68±0.01 | 0.79±0.0 | 0.85±0.01 | 0.78±0.02 (0.81) |  |
| NorESM1 - EF | 0.81±0.00 | 0.7±0.01 | 0.77±0.0 | 0.80±0.02 | 0.78±0.02 (0.82) | 0.96±0.0 (0.97) |

**Table 4.** Intermodel correlations of regional precipitation response for the Samset et al. (2018) models and our models. The average correlation coefficient between the models is 0.34 with a standard deviation of 0.10; the average correlation coefficient between the models used in this study and the Samset et al. (2018) models is 0.38. The correlations are calculated for 50 years with and, 50 years without anthropogenic aerosols. Range of the correlation coefficient shows the standard deviations between results obtained for two different CTRL runs. The correlation for our whole dataset (60+120 years) is shown in brackets.

|  | CESM1 | GISS | HadGEM2 | NorESM1 | ECHAM6-MACSP | NorESM1-MACSP |
|---|---|---|---|---|---|---|
| GISS | 0.38 |  |  |  |  |  |
| HadGEMS2 | 0.42 | 0.43 |  |  |  |  |
| NorESM1 | 0.39 | 0.12 | 0.31 |  |  |  |
| This study |  |  |  |  |  |  |
| ECHAM6-MACSP | 0.42±0.03 | 0.28±0.03 | 0.36±0.03 | 0.12±0.07 |  |  |
| NorESM1-MACSP | 0.5±0.05 | 0.34±0.03 | 0.49±0.0 | 0.38±0.03 | 0.41±0.02 (0.47) |  |
| NorESM1 - EF | 0.54±0.00 | 0.41±0.0 | 0.48±0.0 | 0.33±0.0 | 0.41±0.02 (0.47) | 0.85±0.08 (0.95) |

Tables 3 and 4 show the correlation coefficients for regional climate responses between all experiments in our and Samset et al. (2018) data sets. The correlations are calculated for equilibrium climate runs with equal time averaging over 50 years with and without anthropogenic aerosols both for our and Samset et al. (2018) datasets. Note that these coefficients do not depend on the magnitude of the average responses in the models, but only on the relative regional distributions of the responses. Perhaps surprisingly, the average correlation coefficient for regional temperature response between interactive aerosol models (i.e., the Samset et al. (2018) models), 0.79, is almost identical to the correlation between our prescribed aerosol models (0.78). Also,





the average correlation coefficient between experiments using interactive aerosols and a fully coupled ocean model (Samset et al. models) and experiments using prescribed aerosols and a slab ocean model (our models) is 0.76, nearly the same as for the fully coupled interactive aerosol models only. The similar regional correlation between different experiments is remarkable considering large differences in the aerosol descriptions between the different models. It appears that the differences in aerosol

descriptions do not dominate the differences in regional temperature response.

The average correlation coefficient for regional precipitation changes within Samset et al. (2018) models with intrinsic aerosol descriptions is 0.34, while it is 0.41 within our models with prescribed aerosols. The average correlation coefficient for regional precipitation changes between the Samset et al. (2018) models with fully coupled ocean and our models with a slab ocean is 0.39, which is similar to the mean correlation within the Samset et al. models. The correlation coefficient between

NorESM1 experiments using different aerosol descriptions and ocean models is now only 0.33/0.38. Thus, differences in aerosol descriptions, ocean models and atmospheric responses all contribute to differences in regional precipitation responses. The correlation coefficients for precipitation responses are, however, more uncertain than those for temperature responses, due to a stronger impact of natural variability.

Even long equilibrium climate runs cannot fully eliminate the natural climate variability on a regional level. With our full

dataset (60 of MACSP runs+120 years of CTRL run) we obtain a spatial correlation of 0.47 between NorESM1-MACSP and ECHAM6-MACSP precipitation responses, a slight improvement over the correlation coefficient of $0.41(\pm 0.02)$ for 50+50 year datasets. The spatial correlation for temperature improves from $0.78(\pm 0.02)$ to 0.81. The fully coupled ocean models in the Samset et al. (2018) dataset also feature long-term internal variability in the ocean states that adds to the level of natural variation with respect to our models with simpler slab ocean representations used in this paper. The dependence of the

calculation of time-averaged correlation coefficients on the simulation length for our data is shown in Fig. 5. There, the blue and red shaded region represents the level of expected variation in the regional correlation coefficients between two climate models obtained from equilibrium model experiments with and without anthropogenic aerosols. We obtained correlation coefficient of 0.78 with a standard deviation of $\pm 0.02$ for temperature response and $0.41(\pm 0.02)$ for precipitation after 50 years of simulation, the representative for the Samset experiments but neglecting the impact of long-term ocean variations. The corresponding

correlation coefficients for full model runs (60+120 years of simulation) are 0.47 for precipitation and 0.81 for temperature.

## 4   Conclusions

We have here provided the first results on the equilibrium climate response of modern day anthropogenic aerosols using two different climate models, ECHAM6 and NorESM1, with identical anthropogenic aerosol representations in the models. The results were obtained both using the same representations of aerosol optical properties and cloud-albedo effect and for identical

instantaneous aerosol radiative forcing fields in the models.

The MACv2-SP aerosols produced a very similar total instantaneous anthropogenic aerosol radiative forcing in the two models ($-0.64\,\mathrm{Wm}^{-2}$ in ECHAM6-MACSP and $-0.69\,\mathrm{Wm}^{-2}$ in NorESM1-MACSP experiments). We found that there are





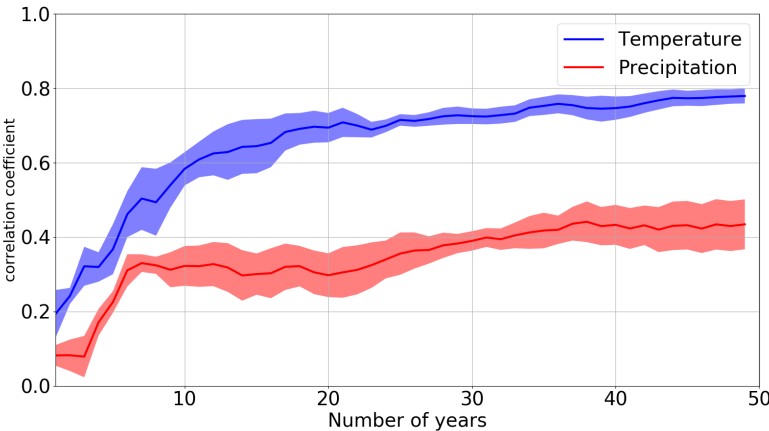

**Figure 5.** Correlation coefficient of temperature and precipitation response as a function of the number of averaged years. Blue is the correlation between the temperature responses to MACv2-SP aerosols in the two models. The shaded area shows the variation between different control runs. The same number of years is used for the CTRL run and MACSP run.

differences up to $3.2\,\mathrm{Wm}^{-2}$ in the instantaneous regional aerosol forcing between the models when using the same aerosol representation. These differences can mostly be explained via differences in cloud fields and surface albedo in the models.

The addition of MACv2-SP anthropogenic aerosols produced very similar global average responses on temperature, $-0.48(\pm0.02)$ K and $-0.50(\pm0.03)$ K, and precipitation, $-1.69(\pm0.04)\%$ and $-1.79(\pm0.05)\%$ in NorESM1-MACSP and ECHAM6-MACSP experiments, respectively. Largest disagreements in regional temperature response were found at high latitude regions associated with largest differences in surface albedo feedback (snow/sea ice), while the largest differences in regional precipitation response were located mainly in the tropics, in part due to changes in the ITCZ. These key regional differences remained even when using exactly the same aerosol radiative forcing fields in both models. Several previous studies have discussed that the main driver for ITCZ shift is the northern hemisphere cooling due to anthropogenic aerosols (Broccoli et al., 2006; Hwang et al., 2013; Wang, 2015). Chiang and Bitz (2005) showed with Community Climate Model version 3 a connection between ITCZ shift and added Arctic ice cover. Based on these previous studies, it seems plausible that different responses in Arctic sea ice and snow cover in ECHAM6-MACSP and in the two NorESM1 experiments result in different high latitude temperature responses, which in turn are reflected as differences in the ITCZ shift that drives the precipitation change at low latitudes. However, it should be noted that the ITCZ shift is also sensitive to the type of ocean model used, and slab ocean models tend to exaggerate the change in ITCZ (Kay et al., 2016).

We compared our results using uniform aerosol representations to a set of four current climate models using their intrinsic aerosol representations but the same aerosol emissions, reported by Samset et al. (2018). Among Samset et al. (2018) models the global responses to additions of anthropogenic aerosol varied between 0.5 K and 1.1 K for temperature and between 1.5% and 3.1% for precipitation. However, the correlation coefficients for regional distributions of climate responses, averaged over




equal run length, were essentially equally good among our experiments with prescribed aerosols and slab ocean representation (0.78 for temperature and 0.41 for precipitation) and among Samset et al. experiments with model-intrinsic aerosols and the fully coupled ocean representation (0.79 for temperature and 0.34 for precipitation). This implies that differences in aerosol descriptions among different models are not the main cause of variation in the regional distributions of climate responses among

5 different models. Rather, differences in model intrinsic dynamic responses appear to dominate the differences in regional climate responses.

Our results imply that in current global climate models the regional aerosol climate impacts cannot be better constrained by further improving aerosol descriptions alone. Improvements on the dynamical cores and physical parameterizations are needed to narrow down model uncertainties in the regional aerosol climate responses.

10 *Data availability.* Data and scripts used for data analysis can be obtained by contacting corresponding author

**Appendix A: Appendix figures A1-A4 and Table A1**

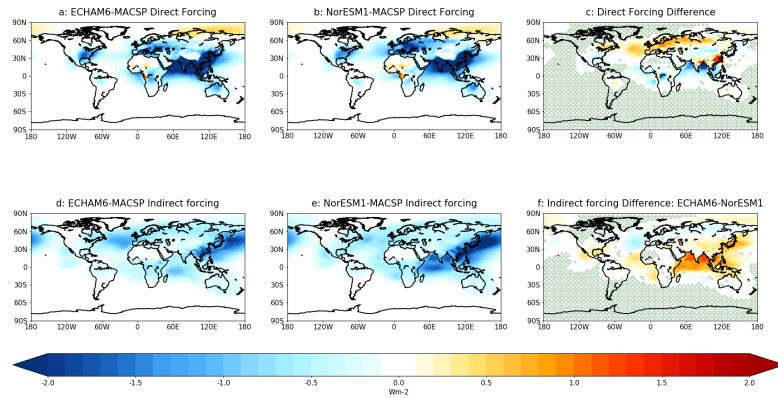

**Figure A1.** Instantaneous radiative forcing by anhtropogenic (MACv2-SP) aerosols. First row shows the direct radiative forcing and second row shows the indirect radiative forcing produced by MACv2-SP. Green masking indicates areas where models do not have a statistically significant difference(ECHAM6-MACSP - NorESM1-MACSP) in instantaneous forcing (p>0.05).





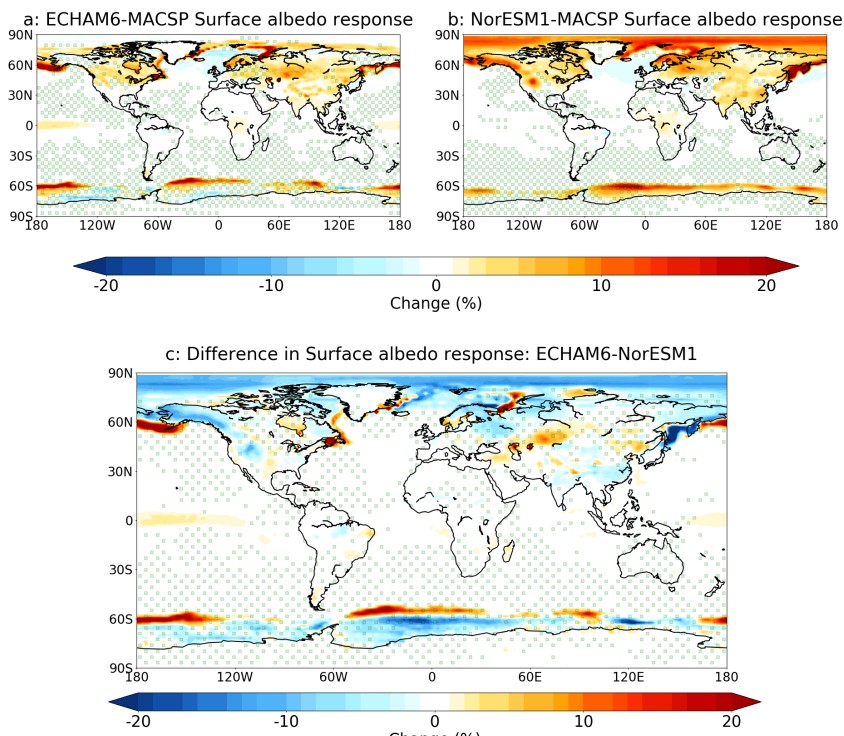

**Figure A2.** Surface albedo response to the addition of anthropogenic aerosols. a: response in ECHAM6-MACSP experiment; b: response in NorESM1-MACSP experiment; c: Difference in surface albedo response: ECHAM6-MACSP experiment minus NorESM1-MACSP experiment. The green dots present the area where aerosols do not have a statistically significant impact at the level $p < 0.05$.



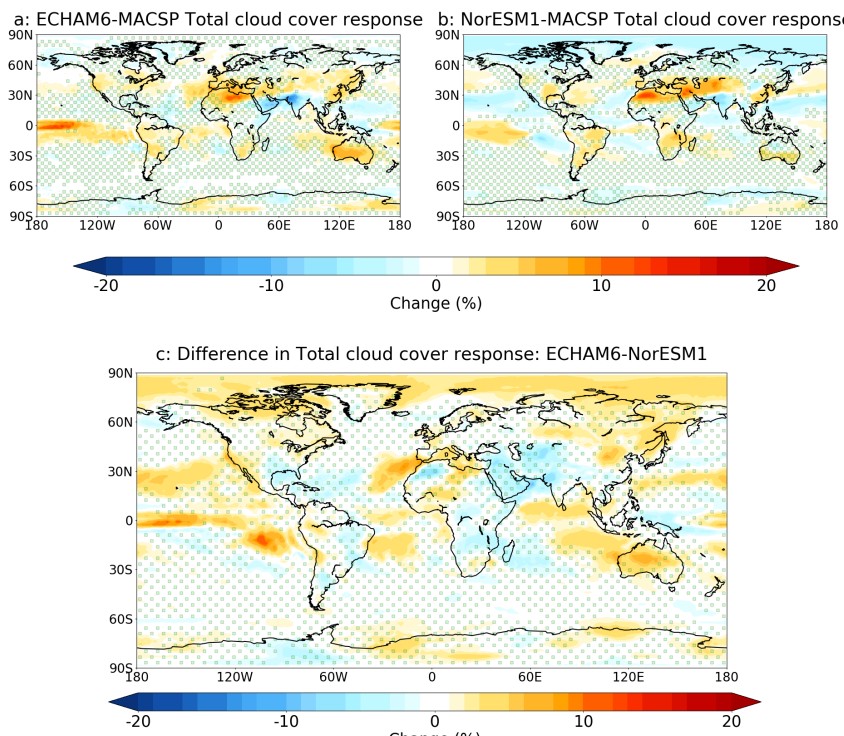

**Figure A3.** Total cloud cover response to the addition of anthropogenic aerosols. a: response in ECHAM6-MACSP experiment; b: response in NorESM1-MACSP experiment; c: the difference in responses between the experiments. The green dots presents area where aerosols do not have a statistically significant impact at the level p < 0.05.



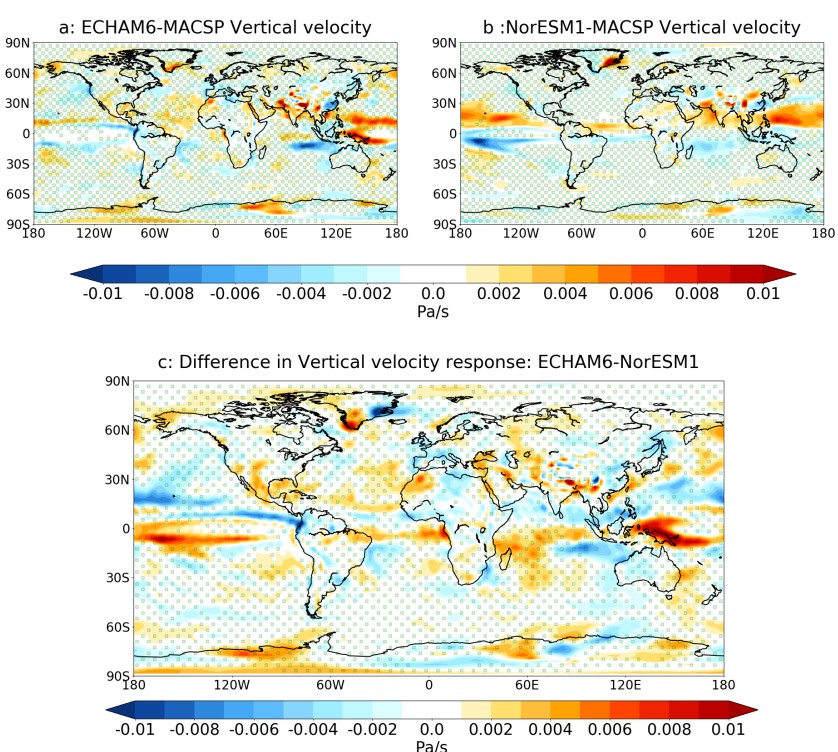

**Figure A4.** Vertical motion response at the 600hPa level to the addition of anthropogenic aerosols. a: response in ECHAM6-MACSP experiment; b: response in NorESM1-MACSP experiment; c: the difference in responses between the experiments. The green dots presents area where aerosols do not have a statistically significant impact at the level p < 0.05.



**Table A1.** Summary of global mean change of temperature and precipitation due to modern day anthropogenic aerosols. Errorbars are standard error of means

| Near surface temperature | | | | | |
|---|---|---|---|---|---|
| | DJF | MAM | JJA | SON | Annual |
| ECHAM6-MACSP | -0.54(±0.03) | -0.50(±0.03) | -0.44(±0.02) | -0.51(±0.02) | -0.50(±0.03) |
| NorESM1-MACSP | -0.49(±0.02) | -0.46(±0.02) | -0.45(±0.01) | -0.51(±0.02) | -0.48(±0.02) |
| NorESM11-EF | -0.51(±0.02) | -0.47(±0.01) | -0.46(±0.01) | -0.50(±0.01) | -0.49(±0.01) |
| Total precipitation (%) | | | | | |
| ECHAM6-MACSP | -1.45(±0.07) | -1.82(±0.07) | -2.11(±0.08) | -1.79(±0.07) | -1.79(±0.05) |
| NorESM1-MACSP | -1.62(±0.07) | -1.53(±0.07) | -2.08(±0.07) | -1.52(±0.06) | -1.69(±0.04) |
| NorESM1-EF | -1.7(±0.05) | -1.68(±0.05) | -2.17(±0.07) | -1.71(±0.04) | -1.82(±0.04) |
| Large scale precipitation (%) | | | | | |
| ECHAM6-MACSP | -1.62(±0.22) | -1.65(±0.12) | -1.22(±0.2) | -0.77(±0.16) | -1.31(±0.1) |
| NorESM1-MACSP | -0.58(±0.21) | -0.83(±0.18) | -2.74(±0.23) | -1.03(±0.16) | -1.28(±0.09) |
| NorESM1-EF | -0.74(±0.18) | -0.98(±0.15) | -2.77(±0.22) | -1.03(±0.09) | -1.37(±0.08) |
| Convective precipitation (%) | | | | | |
| ECHAM6-MACSP | -1.36(±0.12) | -1.91(±0.11) | -2.56(±0.1) | -2.34(±0.1) | -2.05(±0.06) |
| NorESM1-MACSP | -2.27(±0.14) | -1.93(±0.13) | -1.71(±0.11) | -1.82(±0.09) | -1.93(±0.08) |
| NorESM1-EF | -2.28(±0.11) | -2.08(±0.09) | -1.83(±0.08) | -2.12(±0.09) | -2.08(±0.06) |




## Appendix B: Sensitivity analysis of model aerosol forcing

We used a Gaussian process emulation technique (O'Hagan (2006)) to evaluate the regional differences in aerosol radiative forcing. First, we simply assume that the forcing difference depends only on the differences in model output values, and not on the actual values themselves. Second, we selected the differences in modeled output (total cloud cover, surface albedo, precipitation, surface temperature, surface wind u-component) as trial sets for these values. These can be described via a relation $Y = \eta(\boldsymbol{X})$, where $\boldsymbol{X} = [\Delta\alpha, \Delta\beta, ..., \xi]$, where $\alpha$ and $\beta$ are total cloud cover and surface albedo, $\xi$ is pure noise (Gaussian) variable. Next the function $Y = \eta(\boldsymbol{X})$ is inferred using a Gaussian Process prior emulator for a part of the yearly averaged radiative forcing data (in our case, 40 years). Each variable is assigned with a sensitivity index, which describes the relative sensitivity of $Y$ to that variable. The sensitivity analysis of the estimated $Y$ function was done by using Extended Fourier Amplitude Sensitivity Test (FAST) (Saltelli et al. (1999)). As an end result, FAST assesses the contributions of each emulator input variable (components of $X = (X_i)$) to the variance in emulator output variable ($Y$), where it's assumed the input variables $X_i$ have an independent and identical distribution uniform prior. The inferred function $Y$ is finally validated by comparing the emulated forcing field against validation data separate from the training data (here, 20 yearly averaged forcing fields from the model experiments).

*Author contributions.* KN performed ECHAM6 simulations with help from JM,PR and DO. PR performed all NorESM1 simulations. EA adviced on the sensitivity analysis. PU adviced on the data analysis. The manuscript was written by KN and JM, with contributions from all authors. HK came up with the initial research idea and JM coordinated the project.

*Competing interests.* No competing interests are present

*Acknowledgements.* This project has been funded by the European Research Council (ERC) under the European Union's Horizon 2020 research and innovation programme under a grant agreement No 646857, and by the Academy of Finland project 287440. PU was supported by the EC Marie Curie Support Action LAWINE (grant 707262). The author would like also to thank Bjørn Hallvard Samset for providing data of fully couple model runs, and Stephanie Fiedler for providing MACv2-SP code for Echam6.1.





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
