# Peer review of "Role of climate model dynamics in estimated climate responses to anthropogenic aerosols"

_Atmospheric Chemistry and Physics, 2018_

## Referee Comment (RC1) · Anonymous Referee #1 · 21 Feb 2019

The discrepancies in projected climate features among current climate models have recently been related to the differences in representing the processes of aerosol and aerosol-cloud interaction in these models. This study addresses this issue by investigating whether arbitrarily eliminating the differences in models' aerosol forcing strength and distribution could limit the above-mentioned discrepancies. For such a purpose, the authors have designed two sets of equilibrium-climate simulations: firstly to use two climate models (NorESM and ECHAM6) driven by their own slab-ocean modules while masked with the same prescribed direct and first-indirect radiative effects of aerosols (MACv2-SP), and secondly to force one of these two models, NorESM to adopt derived aerosol forcing field from the other model, ECHAM6. Certainly, the results are not entirely a surprise that the two models even with largely the same aerosol forc-

ing distribution and strength would still produce different climate responses particularly over regional scale, for example as reflected from modeled precipitation. The comparison involving the two sets of runs lead the authors to a conclusion that the discrepancy between the two models appears to be largely resulted from the differences in model components beyond that of aerosol.

The model simulations were designed straightforwardly and supportive to addressing the science issue of the study. The paper is well organized, and the result is clearly presented. The content of the paper is perfect for the readers of ACP. Nevertheless, there are a few issues the authors should adequately address before the paper could be accepted for publication.

The authors have drawn one of their major conclusions that "further improvements in the model aerosol descriptions can be expected to have limited value in improving our understanding...". Such a statement does not make any logical sense based on the results of the paper. Firstly, simply making any two models to have a nearly exact radiative effects of aerosols does not necessarily mean that they both had already been equipped with an improved representation of aerosol and aerosol-cloud interaction. Furthermore, we perhaps all agree that such representations in our current climate models are far from being ideal and in fact, even unable to correctly simulate some of the key physical processes. Therefore, no one could predict the outcome in terms of modeled climate features should an ideal aerosol module be eventually produced and included. Secondly, per the current modeling efforts in this study, the applied constraint of aerosol forcing does not even include that on cloud response to aerosol perturbation through precipitation and other critical cloud features – as indicated by the authors, not mentioning that on aerosol resuspension through activation-dissolution-evaporation. Even putting aside these comments relating to rather specific processes, giving the well-known status of our current climate models, logically and realistically, the same conclusion made by the authors to the improvement of aerosols could be applied to any other major model components or aspects. Therefore, the above-commented

statement, especially presented as a major one for the paper, is not logically meaningful and adequate, in addition, it does not accurately reflect the nature and science meaning of this very study.

In order to make a statement as strong as "differences in aerosol descriptions among different models are not the main cause of variation in the regional distributions of climate response among different models", one needs to compare the results produced by the versions of the two adopted models with their own intrinsic aerosol module without arbitrary constraints on forcing. Such a comparison would serve as a good reference to evaluate the real effect by eliminating aerosol forcing discrepancies.

The use of the term "aerosol-cloud interaction" seems quite casual in certain places. Giving the nature of this study that dealing primarily with direct radiative effect alongside the so-called first indirect effect of aerosol, when discussing the context of the study itself, the authors should stay closely within the proper scope of their topic.

The authors borrowed the results presented in Samset et al. (2018) in their discussions. It does tell us that Samset et al. indeed derived a much larger discrepancy among models with intrinsic aerosol scheme. On the other hand, one needs to realize that Samset et al. did not include the same models that are adopted in this study, and the simulation design in that work (with fully-coupled ocean models and most importantly, based on preindustrial era only with perturbations adopted from current climate) are quite different. The performance of climate models with fully coupled ocean component would be different than that of the models with slab ocean module, e.g., likely occurring over high latitudes as discussed in many previous works. The authors should discuss the limitations of such a comparison.

Some specific comments:

Pg. 4, Ln 2: remove "of" before "are based on".

Pg. 4, Ln 5" "between aerosol optical depth and CDNC..." this seems implying that

the modeled AOD rather than aerosol concentration is the primary input for applying MAC-SP constraint? Or, in fact CDNC itself has been prescribed based on MODIS AOD independent to the model predicted aerosol properties?

Section 2.3: The description of NorESM-EF is not very clear. When masking the aerosol forcings of NorESM using ECHAM6 derived values, how did the cloud fields produced in NorESM be considered, for instance, what to do with non-zero first-indirect effect from ECHAM6 in a no-cloud grid in NorESM, or, how to mask direct forcing into cloud fields in NorESM? Could these details be the reason behind the discussed difference between NorESM-EF and ECHAM6-MACSP?

Pg. 12, Line 28: "identical anthropogenic aerosol representations in the models" is inaccurate.

Pg. 14, Ln 1: please correct "essentially equally".

Figure 1, 2, &4: the results of NorESM-EF run should be presented.

---

## Referee Comment (RC2) · Peter Haynes (Referee) · 27 Apr 2019

This paper considers the response of two different climate models to the addition of anthropogenic aerosols. The aerosols are specified in exactly the same form in the two models. The paper argues that whilst the global average temperature and precipitation responses are quite consistent between the two models, there are major differences between the regional responses. The conclusion is that it is the intrinsic differences in the dynamics of the circulation between the two models that determines the differences in the regional responses. This conclusion is supported by additional evidence. The first evidence is from the results of adding the aerosol radiative forcing evaluated in one model into the second – which gives a response more similar to that to the aerosol added to the second model than to that to the aerosol added to the first. The second

evidence is from the response to aerosol in previously reported model experiments which show good spatial correlation, at least in temperature, to the responses in the new experiments reported here.

I see this as an interesting study, which usefully adds to the body of recent work emphasising the importance of the circulation response in determining regional climate change (and also in determining the geographical variation of internal variability that may dominate any clear climate change signal in the short term).

The first referee has already made some comments, which seem reasonable to me, about the general conclusions of this paper – e.g. whether a message of 'the usefulness of research on aerosol representation in models is fundamentally limited until we are more certain about circulation response' is a bit too sweeping.

My own comments are as follows:

General comment: Can you provide any information on the typical geographical distribution of differences in response between the two models you consider for other forcings? Perhaps other results from other experiments with these two models are available (either published or unpublished). Also recent papers on circulation response such as Shepherd (2014, Figure 4) have tended to show differences in winds rather than temperatures. To put your results in context it would therefore be interesting to see differences in, say, 850hPa winds. Also there tends to have been an emphasis on differences in the North Atlantic region. You are showing significant temperature differences in the North Pacific – are you aware of other work that has showed up differences in circulation response between models in that region?

p1 l19: 'unless the dynamical core of the climate models are improved as well'. 'Dynamical core' is often used to mean the part dealing with the dry thermodynamics and dynamics. If that is what you intend then I think that this may be too narrow a view – I don't see why the moist processes, including coupling between circulation and clouds, shouldn't play a role as well.

p3 l16: 'The original ...'

p3 l21: 'were run'

p3 l22: 'intrinsic slab ocean configurations of the models' – again in principle this might be something that controls the different responses of the model and doesn't fit naturally under the heading of 'dynamical core'.

p4 l2: 'properties of are' > 'properties are'?

p4 l10: 'constructed from two identical runs' – do you mean that for each model the control run was constructed from two runs, or do you mean 'two identical runs, one for each model'?

p4 l14: 'experiments' seems an odd word to use about taking differences between simulations.

p5 l3: 'from the MACv2-SP'

p5 l7: The panels in Fig A1 are very small.

p5 l18: 'nearly all the variance' – do you mean 'variance' or 'difference'?

p5 l22: 'Previous research shows that the aerosol radiative forcing can also depend on the meteorology (surface winds and precipitation) produced by the models, partly driven by the natural variability of the climate system (Baker et al., 2015; Bony et al., 2015; Shepherd, 2014).' – my reading of these papers was that they were saying that it was the response to e.g. aerosol radiative forcing, that depends on the meteorology and that the relevant aspects of the meteorology were those that were also responsible for natural variability.

p7 l2: 'disagree the most in high latitude regions' – part of this disagreement is well to the south of 60N.

p7 l3: Are you implying that the changes in surface albedo feedback cause the differ-

ences in temperatures? What reasoning are you using for that?

p7 l6: The disagreement would be 'curious' if the zonal-mean temperature response at high latitudes was locally forced. Are you confident that it is?

p8 Figure 3 etc: You are choosing to quantify the change in precipitation by the percentage change. This means, for example, that in Figure 4 there is a conspicuous difference in precipitation response over much of Australia (where the actual precipitation is very small). I see that in the Samset et al (2017) paper they choose to show change in precipitation normalised by change in temperature. Have you considered carefully whether your choice is the most effective way to show change in precipitation.

p8 l13: 'Africa'

p10 l6: I'm not convinced that the change in vertical velocity can be regarded as a cause of the change in precipitation – don't the two go together as part of an overall coupled change in circulation and precipitation.

p10 l19: For your comparison between NorESM1-MACSP and NorESM1-EF you shown only the zonal-mean and give spatial correlation information. To me the argument that you are trying to make, that the these two simulations closely resemble each other, would be more convincing if you also showed a limit amount of latitude-longitude information – e.g. adding to the information in Figs 2 and 4.

p10 l24: 'dynamical responses' – again I wonder if something like 'circulation responses' might be better (implying something more complex than simply dry dynamics).
* * *

---

## Author Comment (AC1) · 4 Jun 2019

We would like to thank the referee for carefully reading our paper and for the helpful comments and suggestions. We have modified the manuscript according to these suggestions, and detailed answers to each comment are listed below. The reviewer comments are in italic and our answers are in normal font. In the modified manuscript the changes are shown in red font. The modified manuscript can be found from the supplement material of this post

Specific comments:

**Comment 1.**

[Figure]

*The discrepancies in projected climate features among current climate models have recently been related to the differences in representing the processes of aerosol and aerosol-cloud interaction in these models. This study addresses this issue by investigating whether arbitrarily eliminating the differences in models' aerosol forcing strength and distribution could limit the above-mentioned discrepancies. For such a purpose, the authors have designed two sets of equilibrium-climate simulations: firstly to use two climate models (NorESM and ECHAM6) driven by their own slab-ocean modules while masked with the same prescribed direct and first-indirect radiative effects of aerosols (MACv2-SP), and secondly to force one of these two models, NorESM to adopt derived aerosol forcing field from the other model, ECHAM6. Certainly, the results are not entirely a surprise that the two models even with largely the same aerosol foring distribution and strength would still produce different climate responses particularly over regional scale, for example as reflected from modeled precipitation. The comparison involving the two sets of runs lead the authors to a conclusion that the discrepancy between the two models appears to be largely resulted from the differences in model components beyond that of aerosol. The model simulations were designed straightforwardly and supportive to addressing the science issue of the study. The paper is well organized, and the result is clearly presented. The content of the paper is perfect for the readers of ACP. Nevertheless, there are a few issues the authors should adequately address before the paper could be accepted for publication.*

*The authors have drawn one of their major conclusions that "further improvements in the model aerosol descriptions can be expected to have limited value in improving our understanding. . .". Such a statement does not make any logical sense based on the results of the paper. Firstly, simply making any two models to have a nearly exact radiative effects of aerosols does not necessarily mean that they both had already been equipped with an improved representation of aerosol and aerosol-cloud interaction. Furthermore, we perhaps all agree that such representations in our current climate models are far from being ideal and in fact, even unable to correctly simulate some of the key physical processes. Therefore, no one could predict the outcome in*

*terms of modeled climate features should an ideal aerosol module be eventually pro-
duced and included. Secondly, per the current modeling efforts in this study, the applied
constraint of aerosol forcing does not even include that on cloud response to aerosol
perturbation through precipitation and other critical cloud features – as indicated by the
authors, not mentioning that on aerosol resuspension through activation-dissolution-
evaporation. Even putting aside these comments relating to rather specific processes,
giving the well-known status of our current climate models, logically and realistically, the
same conclusion made by the authors to the improvement of aerosols could be applied
to any other major model components or aspects. Therefore, the above-commented
statement, especially presented as a major one for the paper, is not logically mean-
ingful and adequate, in addition, it does not accurately reflect the nature and science
meaning of this very study.*

Author response:
We agree with the reviewer that our conclusion was perhaps overstated, and we fully
agree with the reviewer that aerosol descriptions in current models are far from being
ideal. The root of our original argument stemmed from the fact that models with same
simplified aerosol (or forcing) descriptions (ECHAM6.1 and NorESM1) show no less
regional variability in their climate responses that models with more complex (albeit far
from complete) intrinsic aerosols descriptions (Samset models). However, it is true that
it does not make sense for us to argue that aerosol descriptions would not matter. Nev-
ertheless, even if we would have perfect aerosol descriptions inside the global climate
models, uncertainty arising from the differences in circulation responses between the
models would likely still result in a significant uncertainty in regional climate responses.
For this reason, we have changed our conclusion in the abstract accordingly.

"Hence, further improvements in the model aerosol descriptions can be expected to

have a limited value in improving our understanding of regional aerosol climate impacts, unless the dynamical cores of the climate models are improved as well."

Added:

"Hence, even if we would have perfect aerosol descriptions inside current global climate models, uncertainty arising from the differences in circulation responses between the models would likely still result in a significant uncertainty in regional climate responses"

**Comment 2.**

*In order to make a statement as strong as "differences in aerosol descriptions among different models are not the main cause of variation in the regional distributions of climate response among different models", one needs to compare the results produced by the versions of the two adopted models with their own intrinsic aerosol module without arbitrary constraints on forcing. Such a comparison would serve as a good reference to evaluate the real effect by eliminating aerosol forcing discrepancies.*

Author response:

We agree that this was a too strong statement. In this study, we have not explored regional differences among the same model with different aerosol descriptions, as pointed out by the referee.

Change in the manuscript:

"This implies that differences in aerosol descriptions among different models are not the main cause of variation in the regional distributions of climate responses among

different models. Rather, differences in model intrinsic dynamic responses appear to dominate the differences in regional climate responses."

Added:

"The lack of improvement in the correlation coefficients suggests that differences in aerosol descriptions are not the only cause of regional differences in climate signals between the models. Rather, the differences in model circulation responses appear to dominate the differences in regional climate responses."

**Comment 3.**

*The use of the term "aerosol-cloud interaction" seems quite casual in certain places. Giving the nature of this study that dealing primarily with direct radiative effect alongside the so-called first indirect effect of aerosol, when discussing the context of the study itself, the authors should stay closely within the proper scope of their topic.*

Author Response:

We agree that the use of the term "aerosol-cloud interaction" was too vague particularly in the abstract. First, we have modified the abstract to be more accurate and state specifically only the first indirect cloud effect. Later in the text this term is only used to summarize previous research.

Change in the manuscript:

"We carry out experiments of equilibrium climate response to modern day anthropogenic aerosols using an identical representation of anthropogenic aerosol optical

properties and aerosol-cloud interactions, MACv2-SP, in two independent climate models (NorESM and ECHAM6)"

Added:

"We carry out experiments of equilibrium climate response to modern day anthropogenic aerosols using an identical representation of anthropogenic aerosol optical properties and the first indirect effect of aerosols, MACv2-SP, in two independent climate models (NorESM and ECHAM6)."

**Comment 4.**

*The authors borrowed the results presented in Samset et al. (2018) in their discussions. It does tell us that Samset et al. indeed derived a much larger discrepancy among models with intrinsic aerosol scheme. On the other hand, one needs to realize that Samset et al. did not include the same models that are adopted in this study, and the simulation design in that work (with fully-coupled ocean models and most importantly, based on preindustrial era only with perturbations adopted from current climate) are quite different. The performance of climate models with fully coupled ocean component would be different than that of the models with slab ocean module, e.g., likely occurring over high latitudes as discussed in many previous works. The authors should discuss the limitations of such a comparison.*

Author response:

We agree that our comparison with Samset (2017) dataset has its limitations. However, we would like to point out the Samset et al. study for aerosol reduction is not based on pre-industrial era, but carried out at climate that has warmed by 1.5 K from pre-industrial due to elevated CO2. Samset et.al have also included the same NorESM1 model although with a different ocean description. They have used fully-coupled ocean

models whereas we have slab ocean, and it is known that ocean can play a key role in model discrepancies. It is noteworthy, however, that the spatial correlation coefficients do not differ much between any set of models (Samset et al. models and our models). The role of oceans is now discussed in more detail in the revised MS:

In page 12 we now write:

"The fully coupled ocean models in the Samset et al. (2018) dataset also feature long-term internal variability in the ocean states that adds to the level of natural variation with respect to our models with simpler slab ocean representations used in this paper. Therefore, we would expect the Samset et al. data to include more noise than our results with slab ocean configurations. Furthermore, it is important to note that differences in the ocean descriptions are known to have a large impact in the regional climate responses between different models (Deser et al.; Kay et al. (2016)). Overall, we would expect that due to these differences the climate signals obtained from fully coupled models would intrinsically correlate less well with each other than those from models with slab ocean configurations. Somewhat surprisingly, this turns out not to be the case."

Also we note that we discussed the role of ocean in page 13: "However, it should be noted that the ITCZ shift is also sensitive to the type of ocean model used, and slab ocean models tend to exaggerate the change in ITCZ (Kay et al., 2016). "

**Comment 5.**

*Pg. 4, Ln 2: remove "of" before "are based on".*

Author response:

Done

**Comment 6.**

*Pg. 4, Ln 5" "between aerosol optical depth and CDNC: : :" this seems implying that the modeled AOD rather than aerosol concentration is the primary input for applying MAC-SP constraint? Or, in fact CDNC itself has been prescribed based on MODIS AOD independent to the model predicted aerosol properties?*

Author response:

It is true that this sentence is ambiguous. To be more precise, we have modified the text accordingly:

Change in manuscript:

"The relation between aerosol optical depth and CDNC is derived from MODIS data."

Added:

"The cloud albedo effect in MACv2-SP is parameterized by modifying the model-intrinsic natural cloud droplet number concentration (CDNC) via a relation based on the total change in AOD. This parametrization is derived using MODIS data."

**Comment 7.**

*Section 2.3: The description of NorESM-EF is not very clear. When masking the aerosol forcings of NorESM using ECHAM6 derived values, how did the cloud fields produced in NorESM be considered, for instance, what to do with non-zero first-indirect effect from ECHAM6 in a no-cloud grid in NorESM, or, how to mask direct forcing into cloud fields in NorESM? Could these details be the reason behind the discussed difference between NorESM-EF and ECHAM6-MACSP?*

Author response:

We have now added a section to the appendix of this study to explain more detailed how NorESM-EF run was made, and refer to this section in the main text.

Change in manuscript:

Added into appendix :

The NorESM1-EF run employed radiative forcing extracted from the ECHAM6-MACSP run. First, multi-year monthly means of MACv2-SP aerosol radiative forcing (for TOA and surface radiative fluxes and atmospheric heating rates) were computed for ECHAM6-MACSP. Second, these values were interpolated to the NorESM horizontal and vertical grid and normalized by the monthly-mean incoming solar radiation at model top. Third, during the NorESM1-EF run, these normalized forcings were multiplied by the TOA incoming solar radiation at each radiation time step, and they were added to the radiative fluxes and heating rates computed without MACv2-SP aerosols. This treatment ensures that the diurnal cycle of the aerosol forcing is approximately correct; in particular there is no aerosol forcing during the night. However, the computed forcing is independent of the clouds simulated by NorESM1. Thus, while the aerosol radiative forcing is computed correctly in a monthly-mean sense, its sub-monthly correlation with clouds is ignored. In principle, this could impact the differences between NorESM1-EF and ECHAM6-MACSP. The impact is, however, most likely small. If neglecting the sub-monthly correlation between clouds and aerosol forcing were to have a substantial impact on the climate response to MACv2-SP aerosols, this should also show up in the differences between NorESM1-EF and NorESM1-MACSP. Yet the differences between NorESM1-EF and NorESM1-MACSP are very small (Tables 2 and A1), in fact much smaller than the corresponding differences between ECHAM6-MACSP and either NorESM1-EF or NorESM1-MACSP. This strongly suggests that the differences between NorESM1-EF and ECHAM6-MACSP are primarily caused by the use of a different climate model rather than by the subtle differences in radiative forcing.

**Comment 8.**

*Pg. 12, Line 28: "identical anthropogenic aerosol representations in the models" is inaccurate*

Author response:

This is true as the applied aerosol description results in a different total radiative forcing. We have modified the sentence mentioned in this comment so that it is unambiguous.

The sentence is now changed in the manuscript:

"We have here provided the first results on the equilibrium climate response of modern day anthropogenic aerosols using two different climate models, ECHAM6 and NorESM1, with the MACv2-SP (Stevens et al., 2017) anthropogenic aerosol representations."

**Comment 9.**

*Pg. 14, Ln 1: please correct "essentially equally".*

Author response:

The corrected text now says "nearly as".

Change in manuscript:

However, the correlation coefficients for regional distributions of climate responses, averaged over equal run length, were nearly as good among our experiments with prescribed aerosols and slab ocean representation (0.78 for temperature and 0.41 for precipitation) and among Samset et al. experiments with model-intrinsic aerosols and the fully coupled ocean representation (0.79 for temperature and 0.34 for precipitation).

**Comment 10.**

*Figure 1, 2, 4: the results of NorESM-EF run should be presented.*

Author response:

This is done as suggested. We have included NorESM-EF results for radiative forcing, temperature and precipitation responses, and comparisons with NorESM and ECHAM6 responses.

Change in manuscript:

ECHAM-MACSP and NorESM1-EF difference is added to figure 1.

Figures 2 and 4 NorESM1-MACSP and NorESM1-EF difference are added to show the temperature and precipitation responses for NorESM-EF compared to NorESM1-MACSP.

Please also note the supplement to this comment:

[revised manuscript text omitted]
 | -0.54($\pm 0.03$) | -0.50($\pm 0.03$) | -0.44($\pm 0.02$) | -0.51($\pm 0.02$) | -0.50($\pm 0.03$) |
| NorESM1-MACSP | -0.49($\pm 0.02$) | -0.46($\pm 0.02$) | -0.45($\pm 0.01$) | -0.51($\pm 0.02$) | -0.48($\pm 0.02$) |
| NorESM1-EF | -0.51($\pm 0.02$) | -0.47($\pm 0.01$) | -0.46($\pm 0.01$) | -0.50($\pm 0.01$) | -0.49($\pm 0.01$) |
| Total precipitation (%) | | | | | |
| ECHAM6-MACSP | -1.45($\pm 0.07$) | -1.82($\pm 0.07$) | -2.11($\pm 0.08$) | -1.79($\pm 0.07$) | -1.79($\pm 0.05$) |
| NorESM1-MACSP | -1.62($\pm 0.07$) | -1.53($\pm 0.07$) | -2.08($\pm 0.07$) | -1.52($\pm 0.06$) | -1.69($\pm 0.04$) |
| NorESM1-EF | -1.7($\pm 0.05$) | -1.68($\pm 0.05$) | -2.17($\pm 0.07$) | -1.71($\pm 0.04$) | -1.82($\pm 0.04$) |
| Large scale precipitation (%) | | | | | |
| ECHAM6-MACSP | -1.62($\pm 0.22$) | -1.65($\pm 0.12$) | -1.22($\pm 0.2$) | -0.77($\pm 0.16$) | -1.31($\pm 0.1$) |
| NorESM1-MACSP | -0.58($\pm 0.21$) | -0.83($\pm 0.18$) | -2.74($\pm 0.23$) | -1.03($\pm 0.16$) | -1.28($\pm 0.09$) |
| NorESM1-EF | -0.74($\pm 0.18$) | -0.98($\pm 0.15$) | -2.77($\pm 0.22$) | -1.03($\pm 0.09$) | -1.37($\pm 0.08$) |
| Convective precipitation (%) | | | | | |
| ECHAM6-MACSP | -1.36($\pm 0.12$) | -1.91($\pm 0.11$) | -2.56($\pm 0.1$) | -2.34($\pm 0.1$) | -2.05($\pm 0.06$) |
| NorESM1-MACSP | -2.27($\pm 0.14$) | -1.93($\pm 0.13$) | -1.71($\pm 0.11$) | -1.82($\pm 0.09$) | -1.93($\pm 0.08$) |
| NorESM1-EF | -2.28($\pm 0.11$) | -2.08($\pm 0.09$) | -1.83($\pm 0.08$) | -2.12($\pm 0.09$) | -2.08($\pm 0.06$) |

[revised manuscript text omitted]

---

## Author Comment (AC2) · 5 Jun 2019

We would like to thank the referee for a detailed analysis of our paper. Here we answer to all comments made by referee 2. Importantly, we have changed the term "dynamical response" and fixed typos pointed out by reviewer. Below is a list of our detailed answers to all comments as well as descriptions of the modifications made to the manuscript. In the modified manuscript we have marked all changes with red color. Modified manuscript can be find from supplements of this post.

**Comment 1.**

I see this as an interesting study, which usefully adds to the body of recent work em-

phasising the importance of the circulation response in determining regional climate change (and also in determining the geographical variation of internal variability that may dominate any clear climate change signal in the short term).

The first referee has already made some comments, which seem reasonable to me, about the general conclusions of this paper – e.g. whether a message of 'the usefulness of research on aerosol representation in models is fundamentally limited until we are more certain about circulation response' is a bit too sweeping.

My own comments are as follows: General comment: Can you provide any information on the typical geographical distribution of differences in response between the two models you consider for other forcings? Perhaps other results from other experiments with these two models are available (either published or unpublished). Also recent papers on circulation response such as Shepherd (2014, Figure 4) have tended to show differences in winds rather than temperatures. To put your results in context it would therefore be interesting to see differences in, say, 850hPa winds. Also there tends to have been an emphasis on differences in the North Atlantic region. You are showing significant temperature differences in the North Pacific – are you aware of other work that has showed up differences in circulation response between models in that region?*This paper considers the response of two different climate models to the addition of anthropogenic aerosols. The aerosols are specified in exactly the same form in the two models. The paper argues that whilst the global average temperature and precipitation responses are quite consistent between the two models, there are major differences between the regional responses. The conclusion is that it is the intrinsic differences in the dynamics of the circulation between the two models that determines the differences in the regional responses. This conclusion is supported by additional evidence. The first evidence is from the results of adding the aerosol radiative forcing evaluated in one model into the second – which gives a response more similar to that to the aerosol added to the second model than to that to the aerosol added to the first. The second evidence is from the response to aerosol in previously reported*

*model experiments which show good spatial correlation, at least in temperature, to the responses in the new experiments reported here.*

*I see this as an interesting study, which usefully adds to the body of recent work emphasising the importance of the circulation response in determining regional climate change (and also in determining the geographical variation of internal variability that may dominate any clear climate change signal in the short term).*

*The first referee has already made some comments, which seem reasonable to me, about the general conclusions of this paper – e.g. whether a message of 'the usefulness of research on aerosol representation in models is fundamentally limited until we are more certain about circulation response' is a bit too sweeping.*

*My own comments are as follows: General comment: Can you provide any information on the typical geographical distribution of differences in response between the two models you consider for other forcings? Perhaps other results from other experiments with these two models are available (either published or unpublished). Also recent papers on circulation response such as Shepherd (2014, Figure 4) have tended to show differences in winds rather than temperatures. To put your results in context it would therefore be interesting to see differences in, say, 850hPa winds. Also there tends to have been an emphasis on differences in the North Atlantic region. You are showing significant temperature differences in the North Pacific – are you aware of other work that has showed up differences in circulation response between models in that region?*

*Author response:*

*The reviewer asked for information about the typical geographical distribution of differences in response to different climate forcers between the two models considered in this study. Unfortunately we were not able to find such data. It would have been interesting to compare, for example, responses to heterogeneous aerosol forcing and homogenous greenhouse gas forcing, as done by Shindell et al. (2015). However, as*

*already mentioned, the Shindell et al. paper does not include NorESM1 or ECHAM6 models. However, we note that in future we plan to quantify circulation response differences for a set of different climate forcers within a larger group of models, using PDRMIP data.*

*As suggested by the reviewer, we now also show also 850hPa level wind responses that were discussed by Shepherd (2014, Figure 4) and also recently by Li et al. (2018) (the references are added to the revised manuscript). A figure showing the 850hPa winter wind responses in the two models is now included in the appendix (Shephard (2014) and Li et al. (2018) also discussed wintertime wind responses). We added a short text about the wind response into the manuscript, noting that our results resemble those from the literature.*

*Change in manuscript:*

*Added to page 14:*

*"The lack of improvement in the correlation coefficients suggests that differences in aerosol descriptions are not the only cause of regional differences in climate signals between the models. Rather, the differences in model circulation responses appear to dominate the differences in regional climate responses. Figure C5 shows the average 850 hPa wind responses for ECHAM6-MACSP and NorESM1-MACSP experiments during for Northern hemisphere winter. The responses in the circulation fields vary significantly between the two models, with an annual average correlation coefficient of only 0.18 (DJF:-0.03; MAM:0.07; JJA:0.15; SON:0.19). The lack of robustness in atmospheric circulation responses between different climate models has been previously discussed by Shepherd (2014) for CMIP5 RCP8.5 scenarios and by and by Li et al. (2018) for HAPPI 1.5 K and 2.0 K warming scenarios. Shepherd (2014) argued that the differences in circulation responses cause variation in the regional temperature and precipitation responses in future climate scenarios. Li et al. (2018) showed that model consensus for circulation response is low even for atmosphere-only models forced with*

*same time-varying SST and sea ice, anthropogenic greenhouse gases, ozone, land use, land cover, and aerosols. Both in Shepherd (2014) and Li et al. (2018) data the NH wintertime circulation response over the North Atlantic disagrees significantly between models. Also for ECHAM6-MACSP and NorESM1-MACSP the circulation response over the North Atlantic show differences in magnitude and pattern. Differences are also seen over the North Pacific region. Combined with the difference in the sea ice and surface albedo change in the North Pacific, these circulation changes can drive the temperature response differences in the region.*

**Comment 2.**

*p1 l19: 'unless the dynamical core of the climate models are improved as well'. 'Dynamical core' is often used to mean the part dealing with the dry thermodynamics and dynamics. If that is what you intend then I think that this may be too narrow a view – I don't see why the moist processes, including coupling between circulation and clouds, shouldn't play a role as well.*

*Author response:*

*We thank the reviewer for this insightful comment. We agree that the term dynamical response is too narrow. Therefore, we have change the term, dynamical response, to circulation response as also used in paper by Shepherd, 2014.*

*Change in manuscript:*

*Hence, even if we would have perfect aerosol descriptions inside the global climate models, uncertainty arising from the differences in circulation responses between the models would likely still result in a significant uncertainty in regional climate responses.*

**Comment 3.**

*p3 l16: 'The original . . .*

*Author response:*

*Fixed as suggested*

*Change in manuscript:*

*The Original ECHAM model branched from an early version of the European Centre for Medium-Range Weather Forecasts (ECMWF) model for climate studies.*

**Comment 4.**

*p3 l21: 'were run*

*Author response:*

*Fixed as suggested.*

*Change in manuscript:*

*Here, both models were run with identical fixed modern-day greenhouse gas concentrations*

**Comment 5.**

*p3 l22: 'intrinsic slab ocean configurations of the models' – again in principle this might be something that controls the different responses of the model and doesn't fit naturally under the heading of 'dynamical core'.*

*Author response:*

*This relates also to the comment 2. We have now changed the term dynamical core to circulation response. Also, we have included the oceanic heat exchange as a source for model difference (p3 l22). The role of ocean models is also discussed more in page 12.*

*Change in manuscript page 3 line 22:*

*Oceans were simulated with the intrinsic slab ocean configurations of the models. This idealization removes the effect of natural and aerosol induced variations in ocean circulation and restricts our study to the response in atmospheric circulation, oceanic heat exchange, and sea ice dynamics only.*

*Modified manuscript page 12:*

*"The fully coupled ocean models in the Samset et al. (2018) dataset also feature long-term internal variability in the ocean states that adds to the level of natural variation with respect to our models with simpler slab ocean representations used in this paper. Therefore, we would expect the Samset et al. data to include more noise than our results with slab ocean configurations. Also, it is important to note that differences in the ocean descriptions are known to have a large impact in the regional climate responses between different models (Deser et al (2016).; Kay et al. (2016)). Overall, we would expect that due to these differences the climate signal obtained from fully coupled models would intrinsically correlate less well with each other than those from models with slab ocean configurations. Somewhat surprisingly, this turns out not to be the case"*

**Comment 6.** *p4 l2: 'properties of are' > 'properties are'?*

*Author response:*

*This typo is fixed as suggested by the referee.*

*Change in manuscript:*

*The aerosol properties are based on aerosol climatology by (Kinne et al., 2013).*

**Comment 7.** *p4 l10: 'constructed from two identical runs' – do you mean that for each model the control run was constructed from two runs, or do you mean 'two identical runs, one for each model'?*

*Author response:*

*Our control run is constructed from two almost identical runs via small perturbations on the initial states. The purpose of this approach is to remove some of the climate natural variability by averaging these two runs.*

*Change in manuscript:*

*The sentence "The control run (CTRL) included only natural aerosols, and was constructed from two runs for each model with small initial condition perturbations." is added to page 4*

**Comment 8.** *p4 l14: 'experiments' seems an odd word to use about taking differences between simulations*

*Author response:*

*Word "experiments" was chosen to distinguish individual runs from differences between a pair of runs.*

**Comment 9.**

*p5 l3: 'from the MACv2-SP'*

*Author response:*

*"the" is added here as suggested.*

*Change in manuscript:*

*The total radiative forcing from the MACv2-SP anthropogenic aerosol description was found to be very similar for the two models (see Fig. 1).* **Comment 10.** *p5 l7: The*

*panels in Fig A1 are very small.*

*Author response:*

[Figure]

*We have enlarged this figure and the panels are more visible in the revised MS.*
**Comment 11.**

*p5 l18: 'nearly all the variance' – do you mean 'variance' or 'difference'?*

*Author response:*

*Here we mean variance. This is related to our sensitivity analysis where we used FAST method to decompose the variance in our modelled model difference.*

*Change in manuscript:*

*Our analysis showed that differences in cloud cover and surface albedo can explain nearly all of the variance in the difference in total instantaneous shortwave radiative forcing between ECHAM6 and NorESM1.*

**Comment 12.**

*p5 l22: 'Previous research shows that the aerosol radiative forcing can also depend on the meteorology (surface winds and precipitation) produced by the models, partly driven by the natural variability of the climate system (Baker et al., 2015; Bony et al., 2015; Shepherd, 2014).' – my reading of these papers was that they were saying that it was the response to e.g. aerosol radiative forcing, that depends on the meteorology and that the relevant aspects of the meteorology were those that were also responsible for natural variability.*

*Author response:*

*The reviewer rightly points out that these papers do not discuss the effects of meteo-rology to the aerosol forcing. We have change these references to Fiedler et.al (2019) where they explicitly discuss the effects of model representation of weather to aerosol forcing.*

*Change in the manuscript:*

*Previous research shows that the aerosol radiative forcing can also depend on the meteorology (surface winds and precipitation) produced by the models, partly driven by the natural variability of the climate system (Fiedler et.al, 2019).*

**Comment 13.** *p7 l2: 'disagree the most in high latitude regions' – part of this disagreement is well to the south of 60N*

*Author response:*

*This is true, clearly the models disagree also well to the south of 60N.*

*Change in manuscript:*

*The modeled regional temperature responses between ECHAM6 and NorESM1 simulations disagree the most in mid- and high latitude regions as seen in Figure 2c.* **Comment 14.**

*p7 l3: Are you implying that the changes in surface albedo feedback cause the differences in temperatures? What reasoning are you using for that?*

*Author response:*

*We think that this may indeed be the case, since changes in surface albedo are known to amplify changes in Arctic temperatures (albedo feedback).*

*Change in manuscript:*

*The modeled regional temperature responses between ECHAM6 and NorESM1 simulations disagree the most in mid- and high latitude regions as seen in Figure 2c. In high latitude regions temperature differences are associated with surface albedo responses (snow/sea ice) between the models (see Figure A2). Changes in surface albedo are known to amplify changes in Arctic temperatures (albedo feedback). Hence, differences in snow and sea ice responses may partly explain the difference in temperature responses in the high latitudes.* **Comment 15.**

*p7 l6: The disagreement would be 'curious' if the zonal-mean temperature response at high latitudes was locally forced. Are you confident that it is?*

*Author response:*

*After consideration we decided to remove the entire sentence to which this comment refers to. Particularly, it is not clear what role the changes in cloud cover have on the responses. Also, the point that high latitude responses may not be locally forced (at least fully) is a valid one.*

**Comment 16.**

*p8 Figure 3 etc: You are choosing to quantify the change in precipitation by the percentage change. This means, for example, that in Figure 4 there is a conspicuous difference in precipitation response over much of Australia (where the actual precipitation is very small). I see that in the Samset et al (2017) paper they choose to show change in precipitation normalised by change in temperature. Have you considered carefully whether your choice is the most effective way to show change in precipitation.*

*Author response:*

*We prefer this style of representation and are inclined to keep it as it is. The choice in Samset et al. is, we believe, based on relative large differences in global temperature responses between the models (that do not exist between the two models here).In Samset et al., scaling with temperature was carried out to make the precipitation responses comparable to each other. However, the text in Samset et al. refers to absolute percentage changes.*

**Comment 17.**

*p8 l13: 'Africa'*

*Author response:*

*fixed as suggested by the referee*

*Change in manuscript page 8 line 15:*

*Both models consistently show an overall drying of the Northern Hemisphere, with some statistically significant regional increase in precipitation over the Northwest Africa.*

**Comment 18.** *p10 l6: I'm not convinced that the change in vertical velocity can be regarded as a cause of the change in precipitation – don't the two go together as part of an overall coupled change in circulation and precipitation*

*Author response:*

*It is true that precipitation and vertical velocity go hand in hand. Therefore, the sentence "it cannot be concluded that change in precipitation is caused by the change in vertical velocity" is added to clarify this. Here we want to say that model disagreement in precipitation response is overall related to the difference in circulation response.*

*Change in manuscript:*

*it cannot be concluded that change in precipitation is caused by the change in vertical velocity. Probably, both the changes in vertical velocity and precipitation are related to changes in circulation.*

**Comment 19.** *p10 l19: For your comparison between NorESM1-MACSP and NorESM1-EF you shown only the zonal-mean and give spatial correlation information. To me the argument that you are trying to make, that the these two simulations closely resemble each other, would be more convincing if you also showed a limit amount of latitude-longitude information – e.g. adding to the information in Figs 2 and 4.*

*Author response:*

*Referee 1 also pointed out about the missing information on the NorESM-EF run. We*

*have included the following figures:*

*Change in manuscript:*

*Difference between ECHAM-MACSP and NorESM1-EF added to figure 1.*

*Figures 2 and 4 NorESM1-MACSP and NorESM1-EF difference are added to show the temperature and precipitation responses for NorESM-EF compared to NorESM1-MACSP..*

**Comment 20.**

*p10 l24: 'dynamical responses' – again I wonder if something like 'circulation responses' might be better (implying something more complex than simply dry dynamics).*

*Author response:*

*Word "dynamical" is change to "circulation" as suggested.*

*Please also note the supplement to this comment:*
*https://www.atmos-chem-phys-discuss.net/acp-2018-1335/acp-2018-1335-AC2-supplement.pdf*
* * *
*Interactive comment on Atmos. Chem. Phys. Discuss., https://doi.org/10.5194/acp-2018-1335, 2019.*

**Supplement:**

[revised manuscript text omitted]
 | -0.54($\pm 0.03$) | -0.50($\pm 0.03$) | -0.44($\pm 0.02$) | -0.51($\pm 0.02$) | -0.50($\pm 0.03$) |
| NorESM1-MACSP | -0.49($\pm 0.02$) | -0.46($\pm 0.02$) | -0.45($\pm 0.01$) | -0.51($\pm 0.02$) | -0.48($\pm 0.02$) |
| NorESM1-EF | -0.51($\pm 0.02$) | -0.47($\pm 0.01$) | -0.46($\pm 0.01$) | -0.50($\pm 0.01$) | -0.49($\pm 0.01$) |
| **Total precipitation (%)** | | | | | |
| ECHAM6-MACSP | -1.45($\pm 0.07$) | -1.82($\pm 0.07$) | -2.11($\pm 0.08$) | -1.79($\pm 0.07$) | -1.79($\pm 0.05$) |
| NorESM1-MACSP | -1.62($\pm 0.07$) | -1.53($\pm 0.07$) | -2.08($\pm 0.07$) | -1.52($\pm 0.06$) | -1.69($\pm 0.04$) |
| NorESM1-EF | -1.7($\pm 0.05$) | -1.68($\pm 0.05$) | -2.17($\pm 0.07$) | -1.71($\pm 0.04$) | -1.82($\pm 0.04$) |
| **Large scale precipitation (%)** | | | | | |
| ECHAM6-MACSP | -1.62($\pm 0.22$) | -1.65($\pm 0.12$) | -1.22($\pm 0.2$) | -0.77($\pm 0.16$) | -1.31($\pm 0.1$) |
| NorESM1-MACSP | -0.58($\pm 0.21$) | -0.83($\pm 0.18$) | -2.74($\pm 0.23$) | -1.03($\pm 0.16$) | -1.28($\pm 0.09$) |
| NorESM1-EF | -0.74($\pm 0.18$) | -0.98($\pm 0.15$) | -2.77($\pm 0.22$) | -1.03($\pm 0.09$) | -1.37($\pm 0.08$) |
| **Convective precipitation (%)** | | | | | |
| ECHAM6-MACSP | -1.36($\pm 0.12$) | -1.91($\pm 0.11$) | -2.56($\pm 0.1$) | -2.34($\pm 0.1$) | -2.05($\pm 0.06$) |
| NorESM1-MACSP | -2.27($\pm 0.14$) | -1.93($\pm 0.13$) | -1.71($\pm 0.11$) | -1.82($\pm 0.09$) | -1.93($\pm 0.08$) |
| NorESM1-EF | -2.28($\pm 0.11$) | -2.08($\pm 0.09$) | -1.83($\pm 0.08$) | -2.12($\pm 0.09$) | -2.08($\pm 0.06$) |

[revised manuscript text omitted]